# Coordinate-VAE: Unsupervised clustering and de-noising of peripheral nervous system data

**Hardcastle, Thomas J.**
BIOS
thomas@bios.health

**Lee, Susannah**
BIOS
susie@bios.health

**Wernisch, Lorenz**
BIOS
lorenz@bios.health

**Fortier-Poisson, Pascal**
BIOS
pascal@bios.health

**Shunmugam, Sudha**
BIOS
sudha@bios.health

**Hewage, Kalon**
BIOS
kalon@bios.health

**Edwards, Tris**
BIOS
tris@bios.health

**Armitage, Oliver**
BIOS
founders@bios.health

**Hewage, Emil**[*]
BIOS
founders@bios.health

## Abstract

The peripheral nervous system represents the input/output system for the brain. Cuff electrodes implanted on the peripheral nervous system allow observation and control over this system, however, the data produced by these electrodes have a low signal-to-noise ratio and a complex signal content. In this paper, we consider the analysis of neural data recorded from the vagus nerve in animal models, and develop an unsupervised learner based on convolutional neural networks that is able to simultaneously de-noise and cluster regions of the data by signal content.

## 1 Introduction

Recent advances have made chronic observation [1] of, and limited control [2] over the peripheral nervous system possible. To characterise the dynamics of the signals passing to and from the brain, we wish to categorise patterns of activity within the peripheral nervous system. However, consistent detection of single neuron activity remains a challenge with current cuff electrode technology suitable for *in vivo* neural data acquisition. The relative position of an extracellular recording electrode and neuronal axons close enough to sit above the noise floor affects the polarity of presented signal components [3], and their summation at the electrode occludes the presence of individual action potentials during periods of neuronal activity. Instead, local field potentials (LFPs), the combination of many neuronal responses arriving concurrently at the electrode are observed. These population level responses are potentially informationally richer [4], but preclude the use of conventional spike-sorting [5] methodologies on such data.

Instead, we develop a method based on convolutional neural networks (CNN) that simultaneously de-noises the data and categorises the observed signals. We train this model on approximately one hour of data taken from a single subject approximately twelve hours post surgical implantation. We further show that it is applicable without further training to data from a second subject thirty days post surgical implantation, demonstrating cross-time, cross-subject applicability of the trained models.

## 2 Methods

Neural data are collected from two nerve cuffs implanted on the vagus nerve, each recording LFP data at 30000Hz using a chronically implanted ITX PNS (peripheral nervous system) implant (BIOS,

---

[*]To whom correspondence should be addressed.

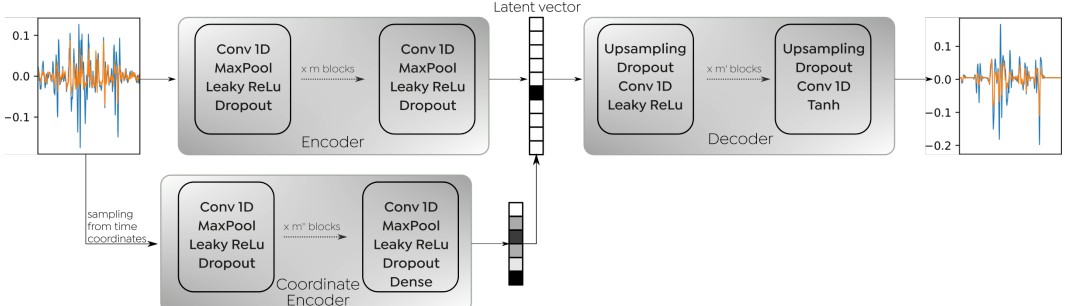

Figure 1: The architecture of the Coordinate-VAE. The input signal is encoded *via* a series of convolutional/pooling/leaky-ReLu/dropout blocks to a categorical latent vector representing the core process observed in the input signal. To allow the decoder to account for phase shifts, time warping, *et cetara*, a set of time coordinates for which the signal is closest to zero are sampled from each channel of the input signal. These pass through a 'coordinate encoder', before being concatenated with the latent vector. The decoder then upsamples with convolution to reconstruct the original signal.

Cambridge, UK). We begin by standardising the mean and standard deviation of the data coming from each cuff, and applying a fifth-order Butterworth bandpass (50-1000Hz) filter, before rescaling such that the training data lie in the range (-1, 1). We then sample small time windows of equal size $w$ from the data as input to the Coordinate-VAE. In the results shown here, $w$ is fixed at 256 samples, that is, at $\frac{256}{30000}$ seconds.

The basic model architecture is shown in Figure 1. For each window, the goal is to reduce the observed data to a one-hot latent vector of size $L$. We achieve this by training a variational auto-encoder (VAE) [6] with a Gumbel-Softmax [7] activation on the latent space. Encoding to the latent space is done through a series of convolutional blocks, where parameters for each block in the encoder are kept constant except for the number of filters in each convolutional layer, which doubles with each block. Pooling takes place in each block where this would not reduce the dimension of the data to less than the size of the convolutional layer. Decoding similarly follows a standard upsampling/convolutional scheme, with a hyperbolic tangent activation following the final convolutional layer. The temperature of the Gumbel layer is slowly annealed throughout training such that the temperature at epoch $E$ is $2e^{-0.0003E}$. During inference, the temperature is fixed at 0.1. We define the loss as a weighted sum of the mean squared error on the reconstruction and the negative of the Kullback–Leibler divergence.

Models were trained in Tensorflow (v 1.12.2) on a single Nvidia Tesla K80. Hyperparameter tuning was carried out over one thousand evaluations for each model using a Tree-structured Parzen Estimator as implemented in the `hyperopt` package [8]. For the primary data set, the data were divided at random into training/validation/test sets comprising 70%, 20% and 10% of the data respectively.

With a small ($L = 20$) one-hot latent space, a standard VAE is unable to reconstruct any signal (Fig. 2(a)). Given a sufficiently large ($L = 50$)) latent space, there is sufficient information to reconstruct the signal, but at the cost of retaining much of the noise, signal artefacts, and increasing the complexity of the latent space (Fig. 2(b)). We solve this by allowing the leakage of some information directly from the original signal to the decoder, bypassing the latent space. For each channel, we find the set of $n$ time-coordinates at which the observed signal is closest to zero. To prevent memorisation of the signal based on these coordinates, we randomly sample a subset $n'$ of these coordinates. Since these data can be ordered over time, we then apply a 1-d convolutional network to this input in a similar fashion to the encoder, giving an encoding of the signal as defined by the sampling from the coordinates. This 'coordinate encoding' is concatenated to the upsampled layers in each step of the decoder. This allows a small ($L = 20$) one-hot latent space to identify the signal present in the data while removing the noise (Fig. 2(c)). For this analysis $n = 5$ and $n' = 1$, that is, a single value taken from the time-axis for each data channel and passed to the encoder *via* a CNN is sufficient to allow reconstruction of the latent space. We are able to reduce the latent space further ($L = 10$ and $L = 5$) while maintaining identification of the signal (data not shown).

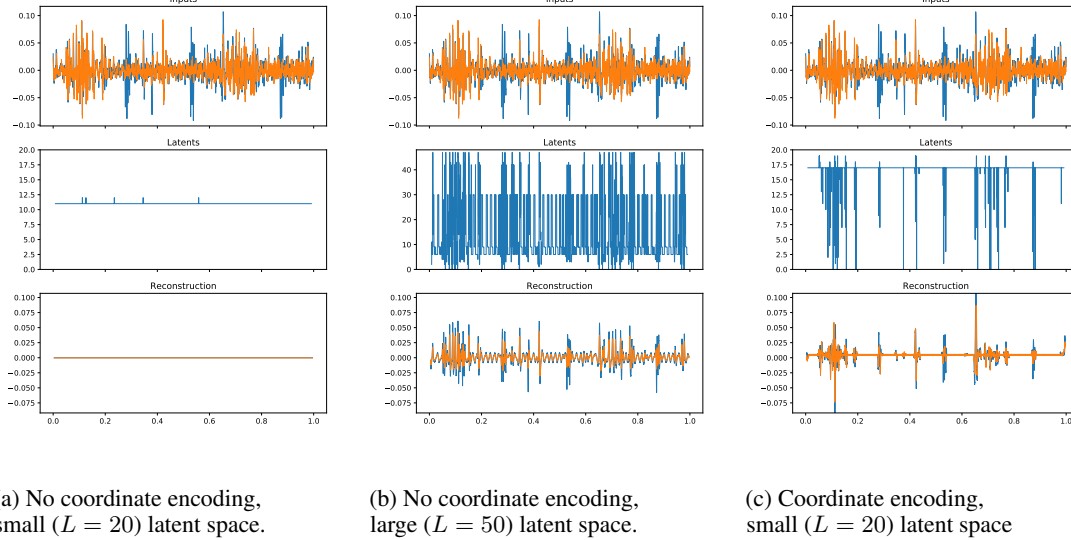

(a) No coordinate encoding, small ($L = 20$) latent space.

(b) No coordinate encoding, large ($L = 50$) latent space.

(c) Coordinate encoding, small ($L = 20$) latent space

Figure 2: Latent space and signal reconstructions inferred from a second of input data from the test set. The input data from the first (blue) and second (orange) cuffs is reduced to a single value in the latent space which evolves over time, and is used to reconstruct the original signal. Without the coordinate encoding, no reconstruction is possible using a small latent space (a). With a large latent space (b), reconstruction is possible but with a complex latent space and the reconstruction of noise in addition to signal. With a coordinate encoder (c) the latent space is relatively simple and the reconstruction is effectively de-noised.

## 3 Results

Figure 2 demonstrates the ability of a Coordinate-VAE model to effectively de-noise peripheral neural data and cluster the observed signals within a relatively simple latent space. Furthermore, we can apply the model trained on data from a single subject to other subjects. Figure 3 shows the latent space and reconstructed signal from vagus nerve recordings from a second subject taken sixty days post surgical implantation. Despite the increased noise levels in this data set, the trained model is able to de-noise the signal and characterise the signals within the data. Data from this subject were human-labelled as containing regions of neural activity corresponding to respiration modulation. There is a clear correlation between respiration modulation events and the amplitude of the reconstructed signal, suggesting that the latent space is able to capture meaningful physiological signals from neural data. Furthermore, the latent space shows strong differences between the latent values prevalent within regions of respiration modulation and those without, with latent values 0, 7, 10, 13, 16 and 19 being significantly ($\chi^2$-test, moderated for dependence of neighbouring values) over-represented within the respiration modulation events. This suggests that, in the absence of labels, the latent space representation may still give useful information with which to identify physiological events.

We explore the de-noising ability of this technique further through simulation studies (Figure 4). We simulate noise in each channel by independently sampling from Morlet wavelets, whose parameters are further sampled from independent normal distributions, and whose location on the time series are uniformly distributed. We combine this 'noise' with 'signal', also sampled from Morlet wavelets, but now located within short 'impulse' time periods and correlated between the two signal channels. We then reconstruct the combined waveform and estimate the ratio of the power of the reconstruction within the impulse regions to the power of the reconstruction outside the impulse regions. By varying the amplitude of the 'noise' signal, we acquire different values for the true signal-to-noise ratio (SNR) and compare this to the SNR post-reconstruction. Particularly for low true SNR, the post-reconstruction data show a considerably improved SNR.

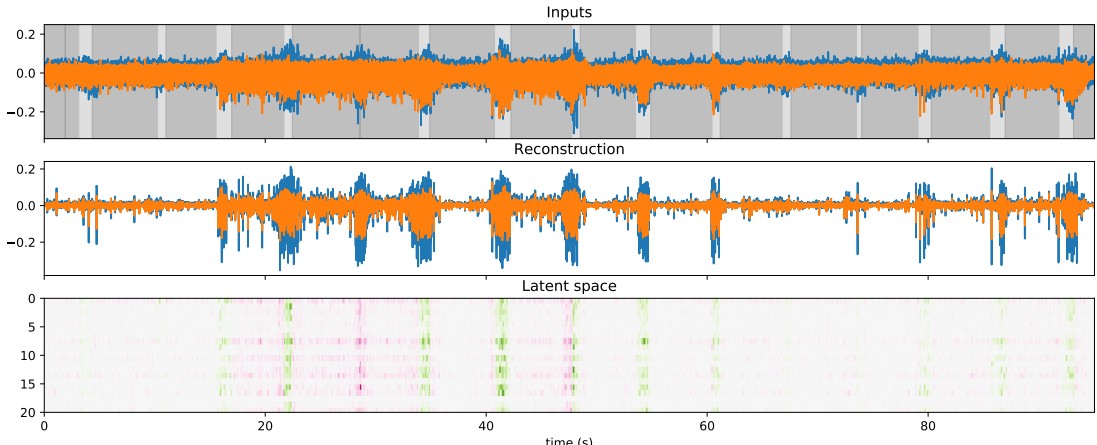

Figure 3: Input data, reconstructed signal, and a smoothed latent space for approximately 100 seconds from a second subject. Data were human-labelled as containing respiration modulation neural activity (light-grey) or not (dark-grey). The reconstructed signal clearly identifies the respiration modulation events. Values for the latent space are shown as a moving mean with a Gaussian kernel with a standard deviation of $\frac{1}{30}$ sec, and are shown as either green (within periods of human-identified respiration modulation) or red (outside those periods), with the intensity indicating the prevalence of a particular latent value. To allow visualisation, the intensities of latent value 17, which indicates the absence of true signal and is by far the most frequent value, have been set to zero in this plot.

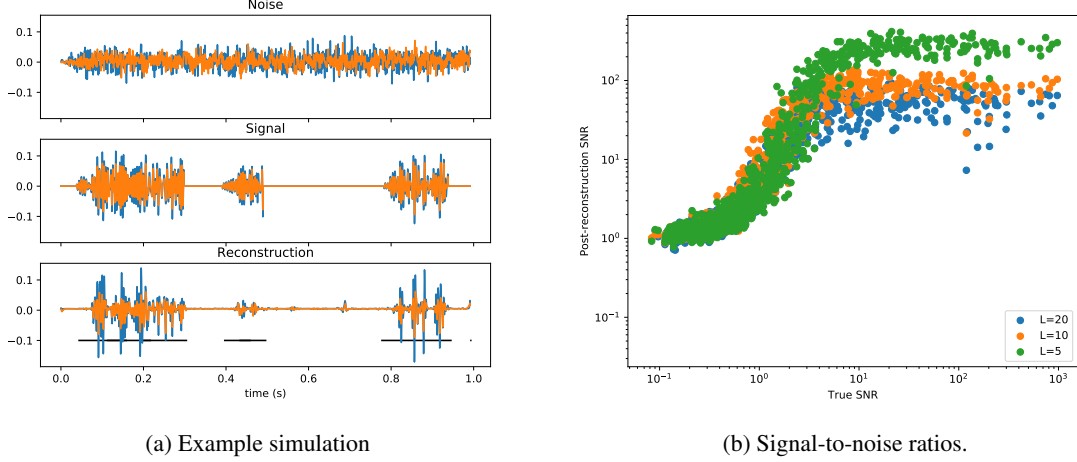

(a) Example simulation

(b) Signal-to-noise ratios.

Figure 4: An example simulation (a), showing noise, true signal, and reconstructed signal. 'Impulse' regions, containing true signal, are indicated by black lines in the reconstruction. SNR for reconstructed signals against true SNR (b) for reconstructions based on Coordinate-VAE with a latent size $L = 20, 10$ and $5$.

## 4    Discussion

The recent development of chronic neural interfacing implant systems that are able to record neural signals over period of months or years will create large sets of primarily unlabelled data, with numerous signals occurring over a range of time-scales. These data are currently un-characterisable with standard methods (e.g. spike-sorting). Previous work in this field has relied on mixing categorical and real-valued latent vectors. Westhuizen *et al* [9] used an adversarial auto-encoder to project neural data to labels, incorporating an approximately one-hot encoding in the latent space but also including an approximately Gaussian vector to allow reconstruction. Since both vectors are trained simultaneously, the Gaussian component of the latent space may contain the relevant labelling information for one

or more true classes. InfoGAN [10], a GAN implementation in which the discriminator identifies components of the latent space is similarly capable of one-hot latent representation of the data, but without constraints on the information carried within the one-hot encoding.

The Coordinate-VAE approach, in restricting the information available to the encoder creating the non-categorical portion of the latent space, allows unsupervised characterisation of the signals in time-series data, while simultaneously de-noising the signal. Models are transferable between individuals, suggesting that we may gain the ability to pre-train large models for the reduction to latent space representations. As shown in Figure 3, there is some evidence to suggest that these latent space representations are also informative for physiological features. We might then rapidly train a final classifier or agent for monitoring or control of individual patients, as in Pandarianth *et al* [11], in which an auto-encoder is used as a dimension reduction technique on collections of neural spiking data acquired from macaque motor and pre-motor cortices, following which a GLM is used to map the complex latent space to spiking activity.

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
