# OpenReview forum: "Coordinate-VAE: Unsupervised clustering and de-noising of peripheral nervous system data"
_NeurIPS.cc/2019/Workshop/Neuro_AI — Real Neurons & Hidden Units @ NeurIPS 2019 Poster_

### Official Review · AnonReviewer1 · 2019-09-25
**Unsupervised VAEs applied to the LFP, potentially compelling but needs more detail in the results**

**Clarity:** 4

**Category:**

AI->Neuro

**Clarity Comment:**

The paper is nicely written for the most part. For the figures, it would be nice to see a zoomed-in comparison between the input signal and reconstruction at a shorter timescale (like 100 ms). For the methods, it says the loss function was MSE and the negative of the KL divergence. Could you clarify what you mean by this? Also, Reference 11 doesn't seem to match its citation.

**Evaluation:**

2: Poor

**Importance:**

4: Very important

**Importance Comment:**

Recent advancements in technology are making it much easier to perform the large-scale recordings of neuronal activity. This "big-data", which can consist of thousands of hours of electrophysiological signals, which can be difficult or otherwise impractical to analyze manually. Developing unsupervised methods that can reduce the dimensionality of this data and cluster together similar states is of critical importance.

**Intersection:**

4: High

**Intersection Comment:**

Research into using unsupervised methods to analyze electrophysiological signals is an excellent example of the applying AI to neuroscience, but it's difficult to evaluate this paper without any additional details on the VAEs performance.

**Rigor Comment:**

There doesn't seem to be any measure or statistics about how much the signal was denoised, besides what is displayed in the figures. Would it be possible to create a synthetic dataset with noise, and report how much the model improves the SNR under increasing levels of noise? Or even adding some noise to the original LFP signal. Additionally, there isn't much info given in regards to clustering. Fig. 3 shows the original output (with human labels) at the top and the reconstruction with labels at the bottom. Are the bottom labels based on some sort of unsupervised clustering of data in the latent dimension? If so, how was the number of clusters determined?

I also have some concerns about the amount of training/testing data and number of subjects. The model was trained on one hour of data from a single subject and then evaluated in a second subject. How much data was used from the second subject? Is it more than the 100 seconds shown in Fig. 3? I think any claims about generalizability would require the use of some sort of statistical measure of performance, followed by evaluations in multiple subjects.

For the VAE, is it possible to train with a smaller latent space? If you reduced the size to a small value (like 2 or 3), could you plot the data and see defined clusters representing respiration states? Also, how is the number of time-coordinates (n) determined, and how does performance change as you change n?

**Technical Rigor:**

1: Not convincing

---

### Official Review · AnonReviewer2 · 2019-09-27
**Promising direction, but key aspects missing.**

**Clarity:** 3

**Comment:**

Definitely needs more work to be able to tell which parts of the model are important for the efficient representation. Promising direction.

**Category:**

AI->Neuro

**Clarity Comment:**

The manuscript is fairly clear, apart from the comments mentioned above. However, when it comes to the method of inputting the coordinate vector, it is quite unclear, even though this is the novelty of the method. It would be good to give an example or provide a figure of the procedure.

**Evaluation:**

3: Good

**Importance:**

3: Important

**Importance Comment:**

The authors consider a specific VAE, i.e. one with additional information going from the input to the output, i.e. the identity of subsampled low-value-time points in the data. Althgouh this approach appears to categorize the data using a small number of latents. It is unclear which parts of the data enable this to work. Simulation results are missing. Some details in the methods are unclear. No intuition is provided as to why the coordinate encoder worked!

**Intersection:**

3: Medium

**Intersection Comment:**

Interesting addition to a common AI model, to denoise neural data.

**Rigor Comment:**

The authors do not mention if their latent space grows with n? Moreover, with the addition of the coordinate encoding, the latent space should be 40 instead of 20, if I understand correctly.

The denoising is interesting, but needs to be compared with simpler methods like low pass filtering and penalized matrix decomposition.

Which parts of the approach are actually important to this technique? The authors have only shown that the coordinate encoder works - which parts of this are important? More importantly, why? Why does this technique work?

Simulation results would be very helpful.

**Technical Rigor:**

3: Convincing

---

### Decision · Program_Chairs · 2019-10-02

Accept (Poster)